# Impact of the Ultrasonic-Assisted Casting of an AlSi7Mg Alloy on T6 Heat Treatment

Inês V. Gomes [1,2,†], José Grilo [1,2,†] , Vitor H. Carneiro [3] and Hélder Puga [1,2,*]

1 CMEMS—UMinho, University of Minho, 4800-058 Guimarães, Portugal
2 LABBELS—Associate Laboratory, 4800-058 Guimarães, Portugal
3 MetRics—UMinho, University of Minho, 4800-058 Guimarães, Portugal
* Correspondence: puga@dem.uminho.pt; Tel.: +351-510-220
† These authors contributed equally to this work.

**Abstract:** In this work, the effect of ultrasonic vibration during solidification on the aging kinetics of an AlSi7Mg alloy is investigated. With the ultrasonic equipment coupled to the mold walls, melt treatment was performed by two approaches: (i) fully above liquidus (>635 °C); and (ii) in the full range between liquidus and solidus (630 °C→ 550 °C). Cast samples were then subjected to T6 heat treatment for different aging times. It is shown that indirect ultrasound treatment increases the cooling rate while active. The eutectic Si was refined and further modified when ultrasound treatment was performed in the semisolid state. Due to the significant release of solute during the decomposition of $\pi$-Al8FeMg3Si6 into fine $\beta$-Al5FeSi, this has a significant impact in the solution stage. Ultrasound treatment fully above liquidus decreased the underaging time to 50% and peak aging time to 25% without compromising strength. The results suggest aging kinetics are correlated with a higher vacancy density and solute enrichment which favors Guinier–Preston (GP) zone formation. These findings show a promising route to tailor the aging kinetics in these alloys by selectively modifying phases and cooling rates.

**Keywords:** AlSi7Mg; ultrasound treatment; intermetallic; solution; aging





## 1. Introduction

Al-Si-based alloys are frequently implemented in numerous industrial sectors, given their physical and mechanical properties as well as cost-effective manufacturing. With an increase in Si content, the castability and fluidity of these alloys are enhanced, and shrinkage effects and melting temperature are reduced [1]. Considering each application (e.g., aerospace, transport, marine etc.), ternary and quaternary alloys are often used to improve the mechanical and material properties of the same binary alloying system [2,3]. Typically, 0.3 wt.% to 4 wt.% of Cu or Mg alloying elements are added to increase the yield strength and hardness, further enhanced by the precipitation hardening of $\theta'$ or $\beta'/\beta''$ metastable phases during heat treatment.

In recent decades, several techniques have been investigated to improve the casting quality to meet the component's mechanical requirements. Ultrasound treatment is among those that are eco-friendly, efficient, and faster. The mechanism of operation is physical, comprising a cavitation effect, which, depending on the thermal properties of the molten melt, promotes the alloy degassing [4], modifies the eutectic Si [5], and refines the $\alpha$-Al grain [6], enhancing the casting sanity compared with other techniques. Although submerged acoustic sonotrodes in the Al molten melt may be the most effective approach for providing ultrasonic vibration, their implementation may have disadvantages. Concerning this approach, Atamanenko et al. [7] stated that the presence of a cold sonotrode or its removal from the molten melt may result in early undercooling. Another issue is the erosion of the Ti sonotrode tip caused by severe cavitation, which shortens the service life of the equipment and introduces residual Ti into the melt [8].

Some authors have investigated different ultrasound (US) treatment techniques before pouring or during solidification. Abramov et al. [9] promoted degassing the molten melt in the ladle (i.e., prior to pouring) and in the molding cavity during the solidification stage, significantly decreasing porosity. Other authors performed ultrasound molten metal treatment through metallic mold/cup using the indirect ultrasonic vibration (IUV) technique. Using IUV to prepare a semisolid slurry of 5083 alloy, Lü et al. [10] showed that cast porosity was reduced with longer ultrasonic vibration periods below liquidus. Khalifa et al. [11] observed that a short direct ultrasound treatment (i.e., sonotrode in contact with the melt) and a prolonged IUV until 580 °C in an AC7A alloy were both capable of grain/secondary phase refining and changing intermetallics. These findings are consistent with our previously published work on Al-Si-Cu and AZ91D alloys [12,13].

According to the ultimate use of the component, a significant quantity of components fabricated by casting alloys of Al-Si-Mg and Al-Si-Cu are often subjected to a heat treatment to further improve their mechanical properties by solid solution and precipitation strengthening. The Al matrix with a higher vacancy concentration is initially saturated with Mg and Si atoms and then quenched to freeze this state under the solution-treated condition. The available vacancy excess allows the precipitation of fine, homogeneous metastable phases, such as $Mg_2Si$, at the subsequent reduced artificial aging temperature [14,15]. Moreover, the eutectic Si modification from platelets to a fibrous morphology promoted during the heat treatment can improve the alloy's intrinsic properties. However, cast components with coarser Si-eutectic morphology require additional time to fragment and fully spheroidize during solution treatment. Extended solution treatments can induce eutectic Si coarsening once spheroidization is completed [3]. Thus, from a logistics and energetic standpoint, there is an interest in optimizing these heat treatments.

In this paper, we study the effect of indirect ultrasound treatment at different stages of solidification on as-cast microstructures and assess its impact on aging kinetics through microstructural and mechanical properties. This study reveals a chance to optimize the mechanical properties by reducing the heat treatment time of castings, which will make the process economically and energetically sustainable compared to the standard approaches.

## 2. Materials and Methods

### 2.1. Experimental Apparatus

Figure 1 depicts the experimental setup used to conduct ultrasonic vibration tests. To conduct the tests, the Multifrequency Multimode Modulation (MMM) technology was used, which consists of a high-power US converter, a booster with an amplification of 1:2.50, a holed waveguide (35 mm in diameter), and a steel mold. The US generator can sweep the frequency to adjust the waveform produced by an MMM ultrasonic generator to produce the appropriate amplitude and largest frequency spectrum in a medium subject to phase changing. The apparatus transfers vibration to the solidifying metal through the waveguide/mold interface in order to provide continuous processing. Each side of the metallic mold was attached to the framework using four compression springs, giving the system more freedom to transfer vibrations to the metal.

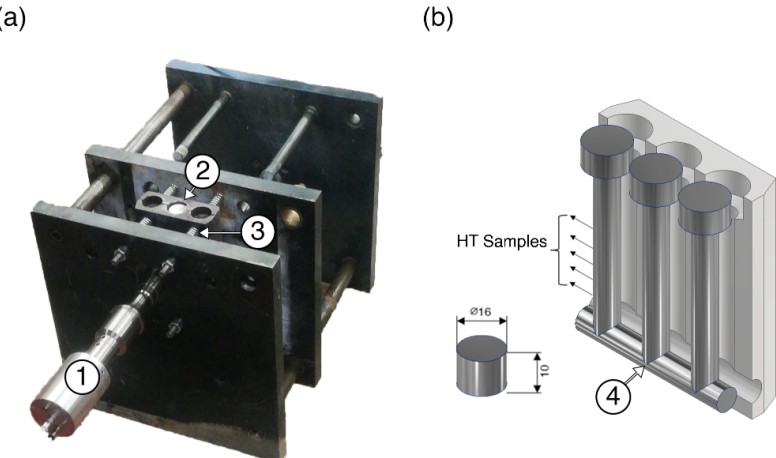

**Figure 1.** (**a**) Experimental setup and (**b**) representation of samples for microstructural and mechanical characterization. (1) Piezoelectric transducer, booster, and waveguide (the last is coupled directly to the exterior mold wall); (2) Steel mold supported by four shafts (3) and compression springs. The thermal behavior was recorded by a thermocouple (4) positioned in the casting's hot area.

### 2.2. Experimental Procedure

AlSi7Mg alloy (2 kg-chemical composition shown in Table 1) was melted in an electrical resistance furnace and homogenized for 15 min in a SiC crucible at $720 \pm 5\,°C$. The molten metal was then allowed to cool to $710 \pm 5\,°C$, degassed with Sialon Tube [16] for 2 min, and poured into a preheated ($250 \pm 5\,°C$) steel metallic mold.

**Table 1.** AlSi7Mg aluminum alloy chemical composition.

| Alloy | Chemical Composition (% wt.) | | | | | | | | |
|---|---|---|---|---|---|---|---|---|---|
| | Si | Mg | Fe | Ti | Cu | Mn | Zn | Res | Al |
| A356 | 7.44 | 0.45 | 0.13 | 0.11 | 0.07 | 0.07 | 0.05 | 0.12 | Bal. |

Ultrasonic processing conditions were designed using the information in [17], namely (i) the liquidus and solidus temperatures of the alloy were, respectively, $614\,°C$ and $554\,°C$; (ii) the formation of primary aluminum dendrites began at $614\,°C$; (iii) the formation of the binary Al-Si eutectic occurred at $574\,°C$; and (iv) the tertiary eutectic and complex intermetallics formed near solidus. Three experimental settings were designed (Table 2) based on the temperature ranges for ultrasonic vibration.

**Table 2.** Processing conditions of the samples.

| Condition | | Sample |
|---|---|---|
| No US Treatment | | A |
| US Treatment Temperatures | From casting temperature to $635 \pm 5\,°C$ | B |
| | From $630 \pm 5\,°C$ to $630 \pm 5\,°C$ | C |

Ultrasonic parameters were set to 500 W of electric power and $20.7 \pm 0.3$ kHz frequency. A LabVIEW-based application using National Instruments CompactDAQ equipment with a thermocouple type K was used to monitor metallic mold temperature and to record temperature–time curves (Figure 1b–detail (4)). A sample from each experiment was kept as-cast state to assess the alloy properties. The remaining samples were solution-heated at $540\,°C$ for 540 min and quenched in room temperature water. Artificial aging was then performed at $170\,°C$ for 8, 64, 128, 256, 360, 480, 720, 1440, and 2520 min.

*2.3. Sample Preparation and Characterization*

Samples were ground with SiC papers up to P4000 and polished with 1 μm polycrystalline diamond suspension, followed by oxide polishing with 0.02 μm colloidal silica. Keller's reagent was used as an etchant for revealing the microstructure of samples for optical microscopy LEICA DM2500 M (Leica Microsystems, Wetzlar, Germany). For the morphological study of the eutectic phase, samples were deep etched for 90 min in 1% HF water solution and examined in a JSM-6010LV (JEOL, Tokyo, Japan) SEM equipped with an energy dispersive spectrometer (EDS) INCAx-act, PentaFET Precision (Oxford Instruments, Abingdon, United Kingdom).

An INSTRON Model 8874 testing machine (INSTRON, Norwood, MA, EUA) was used to perform tensile tests at room temperature with a displacement rate of 0.5 mm/min to determine yield strength and ultimate tensile strength. Each processing condition was evaluated with ten specimens. Officine Galileo Mod. D200 tester (LTF, Antegnate, Italy) with diamond square-based pyramid indenter under 0.5 kgf for 15 s was used for hardness analysis. Each specimen had five measurements taken to compute Vickers hardness's mean and standard deviation.

## 3. Results and Discussion

Figure 2 shows the microstructures of the as-cast samples produced under three different conditions of processing referred to in Table 2. As can be observed, the application of ultrasound treatment promoted the reduction in the grain size from 180 ± 70 μm (sample A) to 90 ± 30 μm (sample B), and 120 ± 40 μm (sample C).

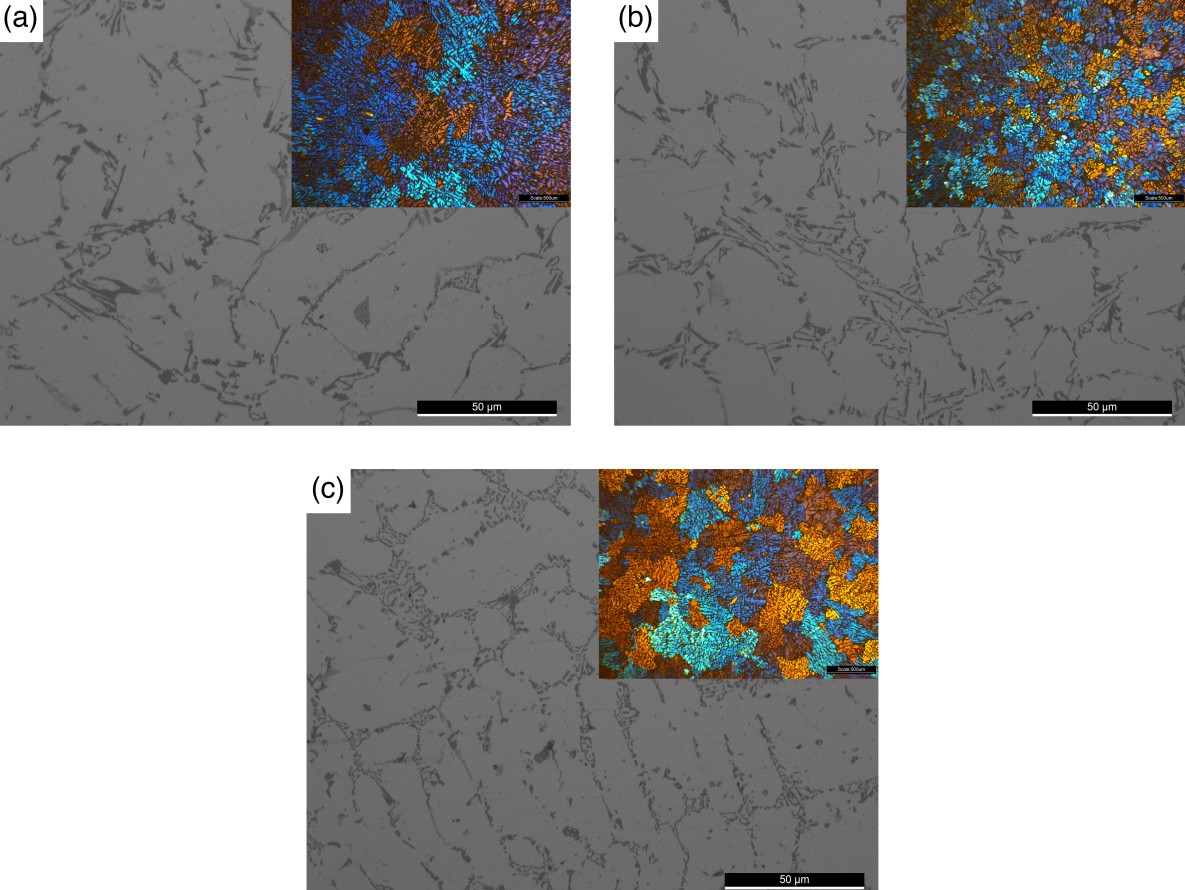

**Figure 2.** As-cast optical and polarized micrographs of (**a**) sample A—no treatment; (**b**) sample B—US treatment fully above liquidus; and (**c**) sample C—US treatment from liquidus to solidus (magnification ×10).

The results depicted in Figure 2 suggest that sample B (i.e., US treatment entirely above liquidus) had a more refined microstructure, which may be associated with faster cooling rates enhanced by acoustic streaming-induced metal stirring generated during the phenomena of cavitation [18,19]. Indeed, when liquid metal is subjected to high-intensity ultrasonic vibrations, the alternating pressure over the cavitation threshold creates small bubbles, which start growing, pulsing in an expansion/compression cycle, and then collapse. During expansion, bubbles absorb melt energy, undercooling the liquid at the bubble–liquid interface and causing nucleation. As bubbles collapse, acoustic streaming occurs in the melt, spreading nuclei into the surrounding liquid and promoting heterogeneous nucleation.

Sample C (US treatment from liquidus to solidus) had a more dendritic microstructure than sample B, as shown in Figure 2c. This evidence can be explained by the fact that when a solid volume fraction is already present in the melt, cavitation can develop acoustic streaming caused by the collapse of bubbles in the remaining liquid, which promotes the fragmentation of already-formed dendrites. This way, the refinement effect becomes less effective at promoting cavitation-enhanced heterogeneous nucleation, a phenomenon typically associated with the first stage of solidification, which was the case with sample B.

As seen in Figure 3, the application of ultrasonic vibration increased the cooling rate of both samples B and C. Even though the sample B cooling rate decreased after the US treatment finished, it remained higher than the non-treated alloy, suggesting that the US treatment may have influenced the eutectic morphology.

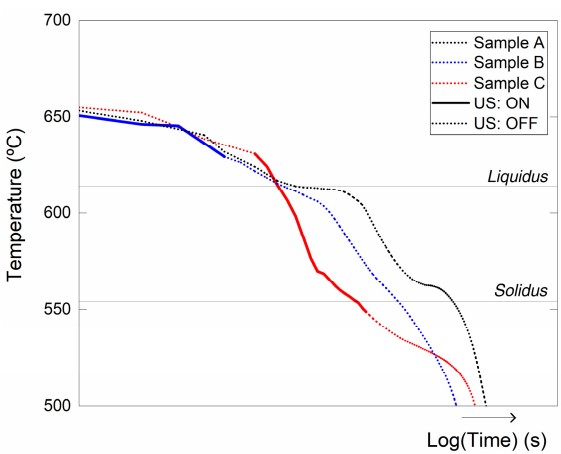

**Figure 3.** Temperature–time for each experimental condition. The active regime of the acoustic energy supplied by the US is denoted by a thicker/wider line.

The morphology of the eutectic Si was observed at a higher magnification using deep-etched samples, as shown in Figure 4.

As seen in Figure 4a,b, sample A exhibited the expected eutectic Si coarse plate shape that commonly grows epitaxially from the $\alpha$-Al dendrites and induces stress concentration that is detrimental to strength and ductility [20]. Even though the shape of the eutectic Si of sample B was similar to sample A, thinner plates were found in the US-treated (sample B) mostly due to the chill modification effect promoted by US treatment.

Figure 4e,f demonstrate that the eutectic Si in sample C presented a rosette-like shape inside a significantly smaller eutectic spacing. It is suggested that US vibration contributed to refining the eutectic Si by promoting the fragmentation of the phase after applying acoustic energy during its formation. In addition, it has been observed that near-equiaxed eutectic Si development occurs when the US maintains a homogeneous solute concentration and temperature field [21]. These hypotheses also explain the absence of considerable modification in eutectic Si when the acoustic energy is applied entirely above liquidus (sample B).

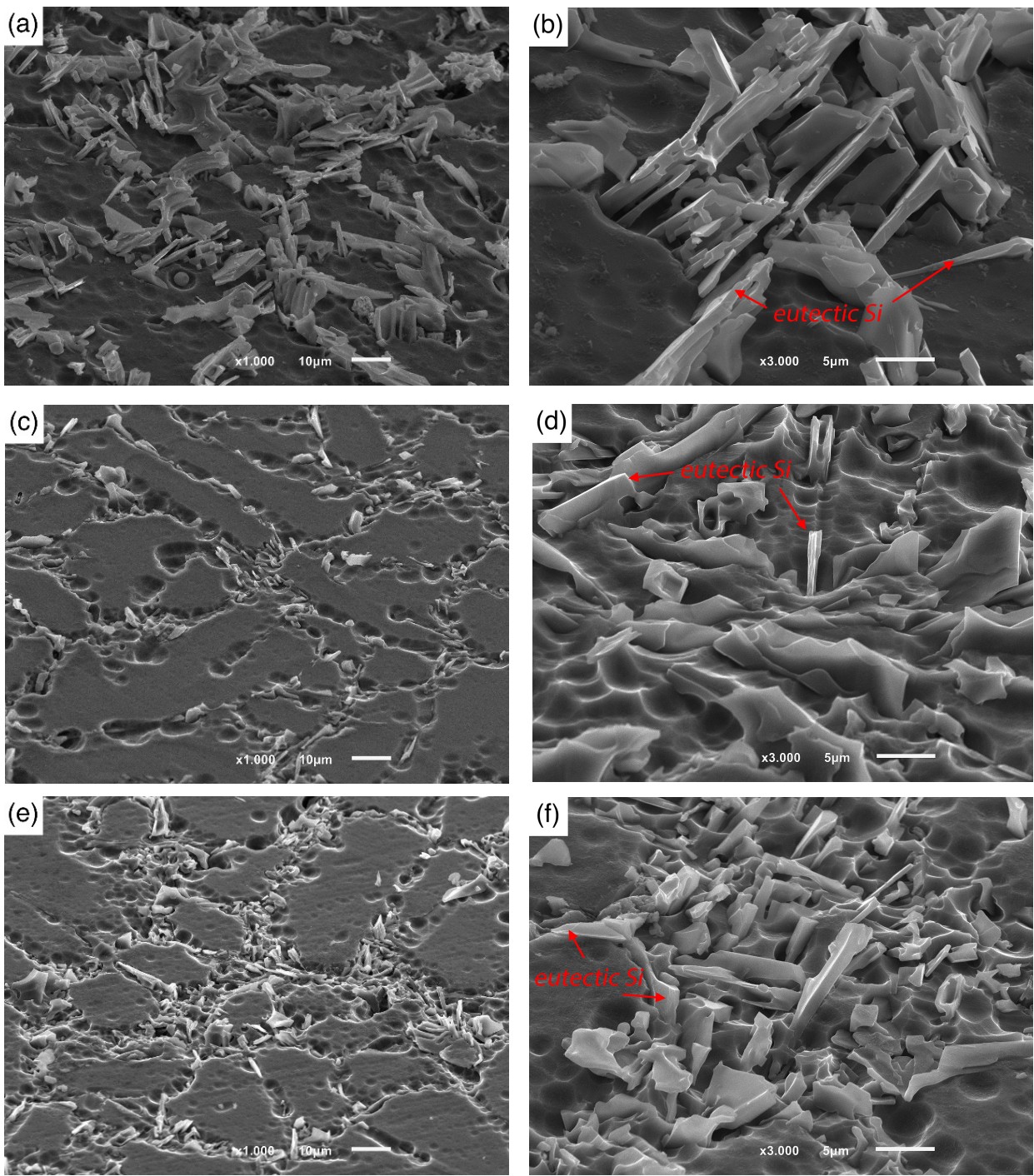

**Figure 4.** SEM micrographs (BSE mode) of the morphology of the fragmented eutectic silicon of (**a**,**b**) sample A; (**c**,**d**) sample B; (**e**,**f**) sample C. (magnification ×1000 and ×3000).

The co-existence of other intermetallic phases, such as $\beta$-Al$_5$FeSi and $\pi$-Al$_8$FeMg$_3$Si$_6$, was confirmed by SEM/EDS analysis, as depicted in Figure 5. EDS analysis suggests the existence of the $\pi$-Al$_8$FeMg$_3$Si$_6$ phase in all the samples, despite the plate-shaped structure in the non-treated sample and the Chinese-script morphology in both US-treated samples. These iron-rich intermetallic compounds ($\beta$-Al$_5$FeSi and $\pi$-Al$_8$FeMg$_3$Si$_6$) result from iron's solubility limit in solid aluminum (0.05 wt.% [22]), which leads it to be rejected from $\alpha$-Al and eutectic and segregated at interdendritic and dendritic edges [23].

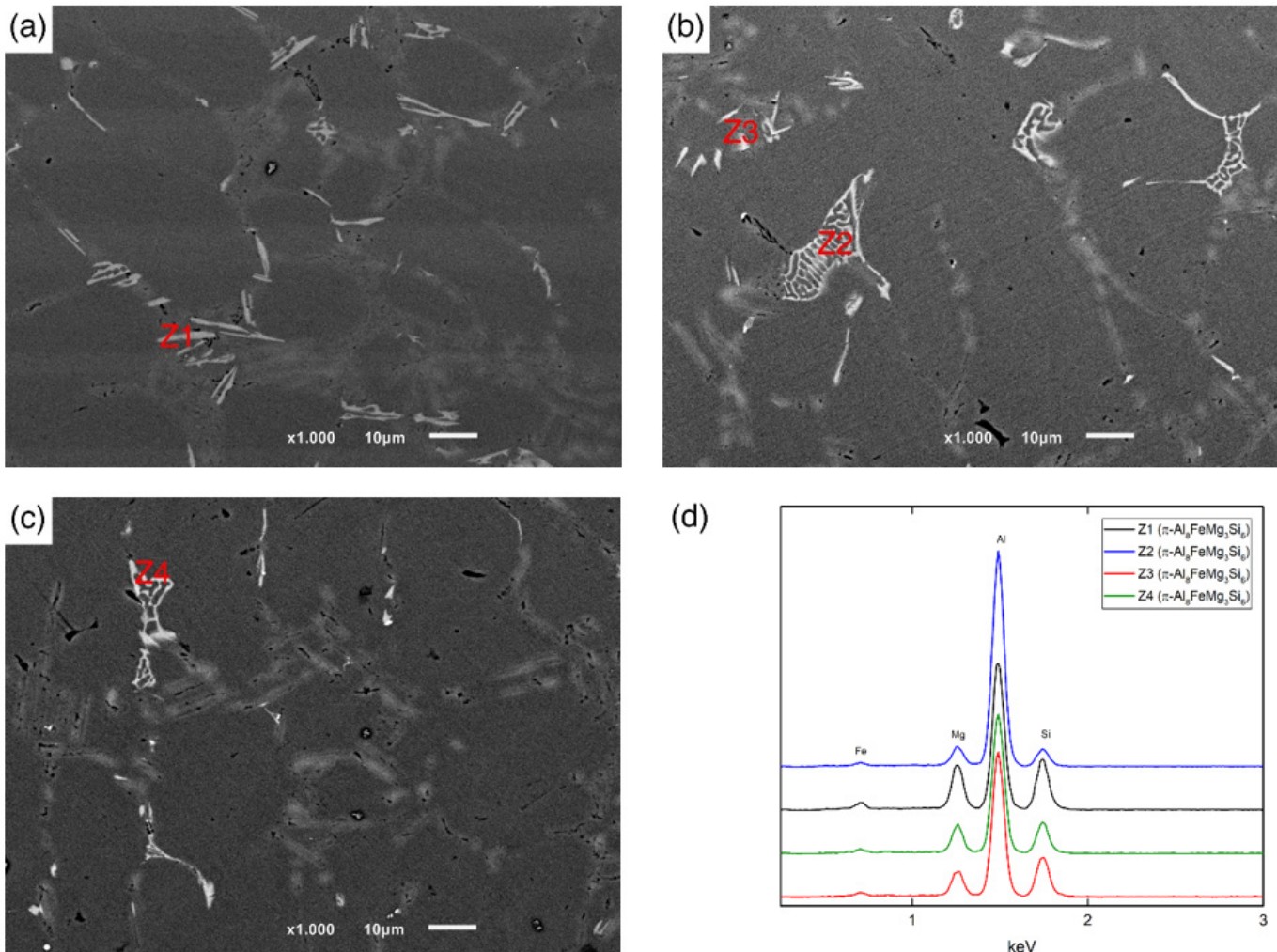

**Figure 5.** SEM micrographs (BSE mode) of the microstructures of samples (**a**) sample A; (**b**) sample B; (**c**) sample C. (**d**) EDS of corresponding intermetallic identified as Z1, Z2, Z3, and Z4, suggesting the overlapping phase composition. (magnification ×1000).

The faster cooling and concentration homogenization promoted by the US treatment may reduce the extent of the solidification reaction responsible for the formation of the $\beta$-$Al_5FeSi$ [24,25] and suppress the $\pi$-$Al_8FeMg_3Si_6$ formation through the peritectic reaction. The proximity of $\pi$-$Al_8FeMg_3Si_6$ to $Mg_2Si$ in the microstructure depicted in Figure 6 suggests that it originated during the final solidification event [26]. Moreover, the intermetallic phase seemed to have a distinct shape in US-treated samples. According to the results, grain refining and solute redistribution under ultrasonic vibration influenced this modification [27]. In sample B, decreasing the cooling rate slowed the eutectic process, enabling iron compounds to settle in the eutectic cell boundaries and forming Chinese-script morphology [28]. In addition, Si eutectic modification may alter the morphology of Fe-rich intermetallic from needle/plate-like to Chinese-script, as already reported by Asghar et al. [28] in comparable observations.

Clearly, the US treatment led to an overall refinement of the microstructure as well as a modification of the eutectic and intermetallic morphology. The AlSi7Mg alloy is typically subjected to T6 heat treatment to induce precipitation hardening in final applications; thus, it is important to analyze the aging kinetics of samples before heat treatment to understand how processing factors affect heat treatment. Figure 7 shows the hardness of solution-treated and artificially aged samples.

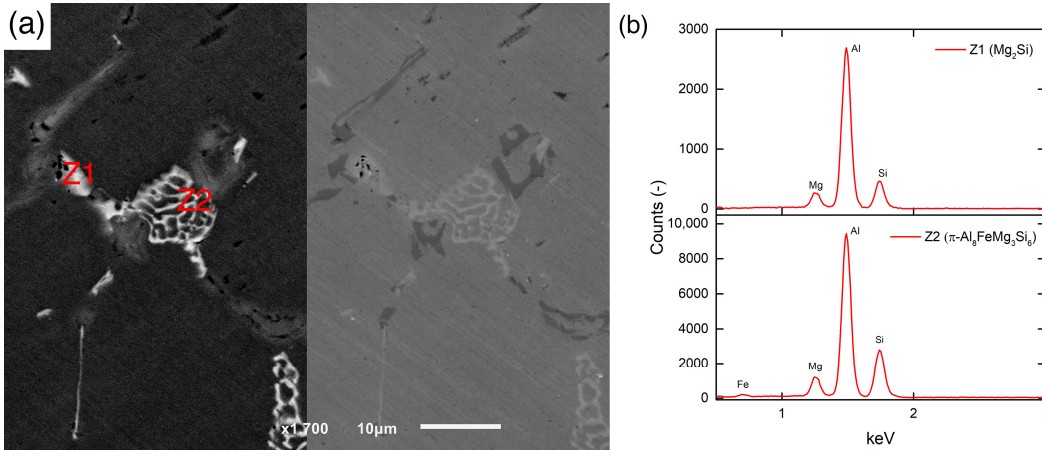

**Figure 6.** (**a**) Detailed SEM (BSE/SE mode) micrograph of Mg2Si (Z1) growing on π-phase (Z2) in sample C (**b**) EDS analysis of corresponding intermetallic identified as Z1 and Z2.

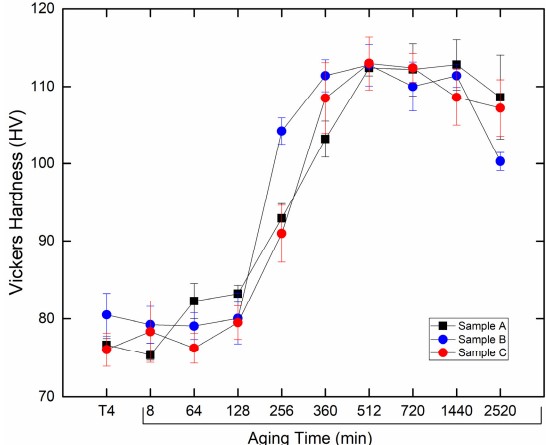

**Figure 7.** Vickers hardness values for solutioned and different artificially aged times.

In the solution-treated condition, sample B showed the highest hardness value comparable to those in samples A and C. This finding may be related to the refinement of eutectic Si, since its initial configuration substantially impacts the thermal modification reaction to heat treatment [29–31]. According to Equation (1), the dissolving time is dependent on the initial size of the silicon particles as well as the supersaturation of the matrix:

$$\frac{dR}{dt} = -\Omega\left[\frac{D}{R_0} + \sqrt{\frac{D}{\pi t}}\right] \tag{1}$$

where $R_0$ is the initial particle size, $\Omega$ is supersaturation, $D$ is diffusion coefficient, and the negative sign denotes R's solution time $t$ gradient. According to the equation, fine particle dissolution rises with solution time [32]. Indeed, higher hardness values are expected for coarser spheroidized eutectic Si.

Since sample C exhibited the most refined as-cast eutectic, it was anticipated that Si spheroidization and diffusion into the Al matrix would occur more rapidly, resulting in an increase in hardness through solution hardening [33]. Nonetheless, it is suspected that the period of the solution treatment exceeded the time required to modify the Si, and therefore the Ostwald ripening process initiated coarsening [34,35].

In addition, sample B's grain size was smaller than the observed in the other samples (Figure 2b), which promoted the interdiffusion process since the distance between eutectic Si and α-Al was shorter. The Hall–Petch strengthening Equations (2) and (3) correlates

the increase in yield strength ($\sigma_y$) and hardness ($HV$) to the decrease in grain size for polycrystalline materials.

$$\sigma_y = \sigma_0 + k_1 D_{GB}^{-\frac{1}{2}} \tag{2}$$

$$HV = H_0 + kD^{-\frac{1}{2}} \tag{3}$$

where $\sigma_y$ is the yield stress, $\sigma_0$ is the initial stress constant, $k$ is the Hall–Petch constant, $D$ is the average grain diameter, $HV$ is the Vickers hardness, and $H_0$ is the initial hardness constant. A finer microstructure corresponds to a greater mechanical response, as shown by Equation (2).

Indeed, the earlier occurrence of peak-aging in sample B was validated by the results of tensile testing (Table 3). It may be observed that peak-aged sample B (i.e., 360 min) displayed similar mechanical properties to the peak-aged sample C (i.e., 512 min) while having a significantly lower artificial aging time. As expected, sample A displayed lower mechanical properties due to the absence of grain/secondary phase refinement.

**Table 3.** Processing conditions of the samples.

|         | Artificial Aging Time | | | |
|---------|:---------:|:---------:|:---------:|:---------:|
| **Samples** | **(360 min)** | | **(512 min)** | |
|         | **YS (MPa)** | **UTS (MPa)** | **YS (MPa)** | **UTS (MPa)** |
| **A**   | -          | -           | $180 \pm 15$ | $210 \pm 12$ |
| **B**   | $214 \pm 9$ | $245 \pm 15$ | $209 \pm 10$ | $248 \pm 7$ |
| **C**   | -          | -           | $220 \pm 16$ | $253 \pm 12$ |

$Mg_2Si$ was also dissolved into the Al matrix during solution treatment conversely to the $\pi$-phase, which has been suggested to take longer [33]. In contrast, Mg and Si from $Mg_2Si$ were dissolved and diffused into the $\alpha$-Al matrix by losing its elemental Mg that was also diffused into the $\alpha$-Al. The solubility limit of Mg governs its diffusion into the $\alpha$-Al, which explains why it was detected even after T6 treatment. In addition, the fraction of the $\pi$-phase in the US-treated samples (i.e., B and C) was lower than in the non-treated samples (i.e., A) in their solution-treated condition. As previously proposed by Kim et al. [30], such findings may be explained by the complete transformation of the fine $\pi$-phases, which were more prevalent in US-treated samples. Figure 8 further demonstrates that the intermetallic fraction was similar across non- and US-treated samples, but the area of its particles decreased compared to the non-treated samples. The $\pi$-phases particles of the non-treated (sample A) exhibited a necked and fragmented morphology, while in the US-treated samples, their morphology was closer to spheroidization. Sample C was significantly more fragmented and transitioned into round particles more than sample B.

Up to 128 min of artificial aging, the hardness of these samples exhibited a similar trend; thus, it is thought that solid solution hardening was the primary strengthening mechanism. The variation in hardness is related to the previously detailed microstructural changes and the migration of solute atoms into matrix vacancies [36]. However, when the samples were artificially aged for 256 min, significant hardness variation was seen between samples B, A and C, suggesting that solute atoms already formed clusters and precipitation phenomena may have started. According to Q. G. Wang and C. J. Davidson [28], dendritic arm spacing or Si modification does not significantly affect Mg-induced precipitation, suggesting that microstructure's effect on the precipitation process is restricted if Mg concentration is unaffected. Therefore, the remaining precipitation-influencing characteristics may be linked to solute and vacancy distribution. Thus, solution hardening gradually lost prominence while coherence hardness emerged as the primary strengthening mechanism, as suggested in Figure 9.

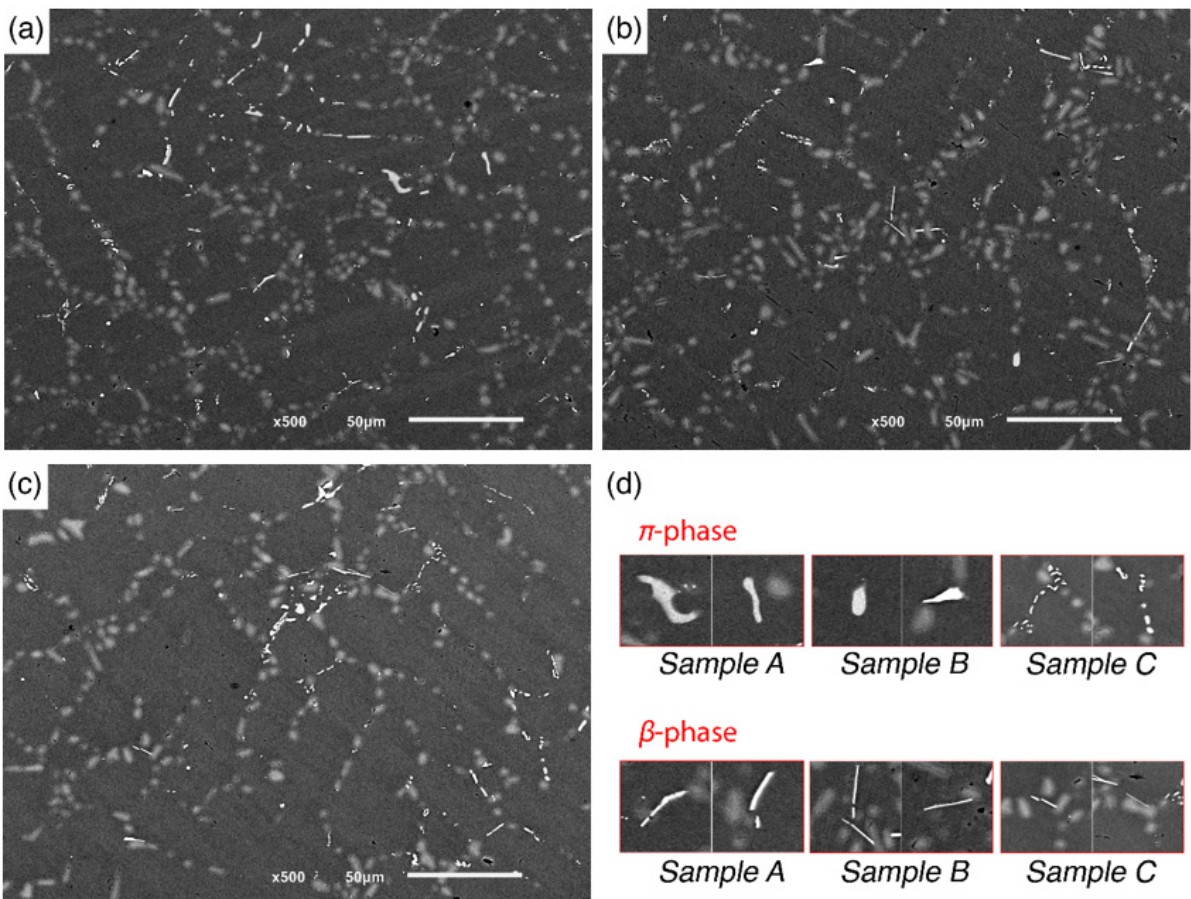

**Figure 8.** Detailed SEM (BSE mode) micrograph of solution-treated microstructures of (**a**) sample A, (**b**) sample B, (**c**) sample C and (**d**) highlighted intermetallic for each sample.

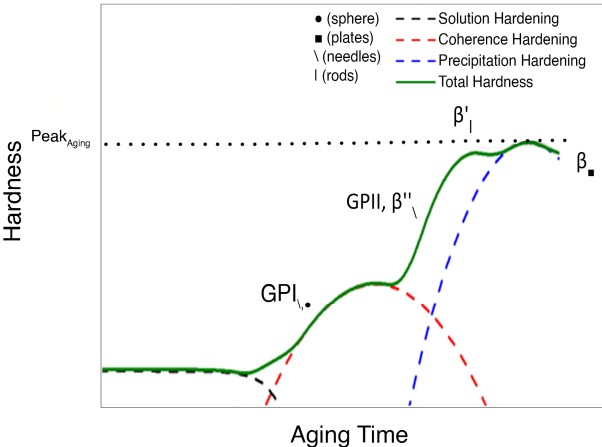

**Figure 9.** Theoretical representation of the hardness evolution as a function of the artificial aging time and the associated physical mechanisms of samples A, B, and C.

Increasing the cooling rate may limit vacancy annihilation, retaining a vacancy-richer supersaturated solid solution where GPI zones start nucleating. The homogeneous distribution of vacancies may lower the vacancy–solute diffusion distance, accelerating precipitation–strengthening mechanisms. Moreover, a significantly increased number of defects, such as vacancies, has been pointed out previously due to metal sonication [29] and heat treatment studies [30].

Acoustic energy in its liquid state may result in a more homogenous solute and vacancy distribution in the melt, meeting the stoichiometric relation necessary for GPI zone nucleation, resulting in the sample B reaction. Enhanced solute dispersion via the US was still in effect above liquidus and semisolid (sample C), but diffusion became more difficult as the solid fraction continues. This may explain why samples A and C took longer to reach the same hardness as sample B at 256 min. If a solute-enriched area is evenly distributed, numerous GP zones may act as precipitation nuclei in a short time.

GPI zones are entirely coherent with the Al matrix and as they are metastable, they disintegrate in the presence of a more stable phase, according to the predicted precipitation sequence illustrated in Figure 9, following Edwards et al. [31] and Vissers et al. [32]. Artificial aging for 512 min produced equivalent hardness values for all samples, showing that microstructure and composition in bulk were comparable. For samples with the same Mg fraction available for precipitation, melt treatments were not expected to improve the alloy peak aging since the matrix cannot retain more solute in the solution-treated condiion [28]. According to previous studies [33], $\beta''$ is the main strengthening phase, so it may be assumed that the microstructure at peak aged condition is mainly composed by this phase regardless of the processing conditions. As the aging treatment proceeds, the precipitates transform into the transition phases $\beta''$ and $\beta'$, progressively losing coherency with the matrix and leading to that precipitation hardening effect surpassing that of coherence hardening.

The growth of the precipitates is beneficial for the materials' strength as long as they may be sheared by dislocations (i.e., Friedel effect), increasing its hardness and strength as described by Equation (4) [34]. However, in the overaged condition, all the precipitates are already formed, the coherency is fully lost, and non-shearable $\beta'$ and $\beta$ (Mg$_2$Si) become coarser by Ostvald ripening [35]. Consequently, the Orowan bowing stress decreases and the dislocation mechanism is shifted to an Orowan mechanism. According to Equation (5), if precipitates surpass the critical radii, hardness and strength will decrease for longer aging periods [36].

$$\sigma_{PPT} = c_i f_r^{\frac{1}{2}} r^{\frac{1}{2}} \rightarrow r < r_{trans} \tag{4}$$

$$\sigma_{PPT} = c_i f_r^{\frac{1}{2}} r^{-1} \rightarrow r > r_{trans} \tag{5}$$

where $\sigma_{PPT}$ is the precipitation strengthening, $c_i$ a constant (dependent on the Taylor factor, Burgers vector, precipitate shear modulus and interfacial energy), $f_r$ the volume fraction of the precipitates, $r$ the radii of the precipitate, and $r_{trans}$ is the transition radius.

## 4. Conclusions

The presented study detailed the effect of the US treatment applied at different stages of AlSi7Mg solidification on the artificial aging kinetics, by correlating the resultant microstructural morphologies with their hardness and tensile strength. The following main conclusions were drawn:

1.  Despite its short vibration time, ultrasound treatment enhanced the heat transfer of the alloy/mold combination during cast filling and solidification. US treatment displayed grain refinement and eutectic Si modification regardless of the solidification stage;
2.  The sample with ultrasound treatment entirely above liquidus (i.e., > 635 °C) showed the refined grains relatively to the other samples. Even though the samples whose treatment was performed through liquidus to solidus (i.e., 630 °C → 550 °C) showed grain refinement, especially when compared with non-treated samples, it was still less efficient;
3.  The modification of eutectic Si was more efficient when the ultrasound was active during the eutectic nucleation temperature, as in the case of samples whose treatment was performed from liquidus to solidus (i.e., 630 °C → 550 °C);
4.  The morphology of $\pi$-Al$_8$FeMg$_3$Si$_6$ phases changed from coarse plates in non-treated samples to a Chinese-script morphology in samples whose ultrasound treatment was

performed fully above liquidus (i.e., > 635 °C). Both coarse and Chinese-script morphologies were observed when treatment was imposed through liquidus to solidus. After solution treatment, all the samples exhibited spheroidized Si and evidence of $\pi$-Al$_8$FeMg$_3$Si$_6$ transformation into $\beta$-Al$_5$FeSi;

5.　The peak aging condition occurred firstly when ultrasound treatment was performed fully above liquidus (360 min). The aging time was significantly reduced relatively to the other samples (512 min). The accelerated aging kinetics were attributed to the difference in vacancies density and solute concentration due to the processing conditions: imposing acoustic vibration entirely above liquidus promoted the solidification of a vacancy-richer matrix, which favored the formation of GP zones. Consequently, the overall aging kinetics were enhanced, and peak aging time was reduced.

**Author Contributions:** Conceptualization, I.V.G., J.G. and H.P.; Methodology, I.V.G., J.G. and H.P.; Validation, V.H.C. and H.P.; Formal Analysis, I.V.G. and J.G.; Investigation, I.V.G., J.G. and V.H.C.; Resources, H.P.; Data Curation, J.G. and V.H.C.; writing—original draft preparation, I.V.G., J.G. and H.P.; writing—review and editing, V.H.C. and H.P.; Supervision, H.P.; Funding acquisition, H.P. All authors have read and agreed to the published version of the manuscript.

**Funding:** This work was supported by PTDC/EMEEME/30967/2017 and NORTE-0145-FEDER-030967, co-financed by the European Regional Development Fund (ERDF), through the Operational Programme for Competitiveness and Internationalization (COMPETE 2020), under Portugal 2020, and by the Fundação para a Ciência e a Tecnologia–FCT IPP national funds. Moreover, this work was supported by Portuguese FCT under the project UIDB/04436/2020 and the doctoral grants PD/BD/140094/2018 and 2020.08564.BD.

**Data Availability Statement:** Not applicable.

**Conflicts of Interest:** The authors declare that there is no conflict of interest.

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
