# Peer review of "Impact of the Ultrasonic-Assisted Casting of an AlSi7Mg Alloy on T6 Heat Treatment"

_metals, doi:10.3390/met13020255_

Round 1
Reviewer 1 Report
Line 41: “a-Al grain”. It has to say α-Al grain
Table 2: US-treatment (start): B: “before pouring”. It starts after pouring.
Line 126-131: It must be explained how many samples were used for tensile tests at each condition
Line 136-145: It is not clear why, in sample B, the cooling rate coincides with sample C during the US treatment. The cooling rate (in B) is faster once the treatment has stopped and the liquidus line is crossed.
Figure 3: the scale time (X axe) must have information about the time. It is not clear if it refers to seconds, minutes and how many.
Figure 3: I think that the change in the slope produced between solidus and liquidus lines is interesting. This change is subtle in the B sample, noticeable in C sample and big in the A sample. It occurs at about 575oC, the temperature of eutectic and intermetallic formations. Some reflections about this slope could be given.
Figure 3: It is not clear the reason of the fast cooling rate of sample C compared with B and C,
Figure 4: I think that the eutectic of the B sample is narrower than the eutectic in A and C (although in the C sample the plates are thinner). This could be because molten is more homogeneous and therefore the α-Al will be more saturated. That could explain the higher hardness in T4 and the fast increase in hardness in T6.
Table 3: Missing results for A and C samples (360 minutes aging time)
Table 3: UTS value for sample A for 512 min (210MPa) is lower than the values for B and C. However, for 512 min hardness values of A, B and C are the same. An explanation should be given
Line: 267-268: “In addition..... than in the non-treated samples” something is lost in this sentence.
Figure 8 (d): the π-phase for the sample C appears small and fragmented but in the © image of the Fig. 8 (center upside) there is a fragmented particle and in the center of the same image there is a big one.
Author Response
See attach.

Reviewer 2 Report
The authors investigated the effect of ultrasonic vibration during solidification on the ageing kinetics 12 of an AlSi7Mg alloy. The study on the effect of ultrasound on the semisolid state alloy is of great interest.
1. Please explain the meaning of GP and GPI.
2. The layout of Table 2 is not clear enough. It needs to be improved.
3. Can the authors provide some statistical data, such as the average grain size and the number of grains.
4. Please mark the position of the eutectic Si and the thinner plate in Figure 4.
Author Response
See attach.

Round 2
Reviewer 1 Report
Thank you for your answers. I do not have more comments.